# Malnutrition Patterns in Children with Chronic Kidney Disease

**DOI:** 10.3390/life13030713

**Published:** 2023-03-06

**Authors:** Vasiliki Karava, John Dotis, Antonia Kondou, Nikoleta Printza

**Affiliations:** Pediatric Nephrology Unit, 1st Department of Pediatrics, Hippokratio General Hospital, Aristotle University of Thessaloniki, 54642 Thessaloniki, Greece

**Keywords:** protein energy wasting, obesity, sarcopenia, sarcopenic obesity, frailty, muscle wasting, muscle strength, abdominal obesity, growth hormone, physical activity

## Abstract

Malnutrition is frequent in children with chronic kidney disease (CKD). Apart from undernutrition and protein energy wasting (PEW), overnutrition prevalence is rising, resulting in fat mass accumulation. Sedentary behavior and unbalanced diet are the most important causal factors. Both underweight and obesity are linked to adverse outcomes regarding renal function, cardiometabolic risk and mortality rate. Muscle wasting is the cornerstone finding of PEW, preceding fat loss and may lead to fatigue, musculoskeletal decline and frailty. In addition, clinical data emphasize the growing occurrence of muscle mass and strength deficits in patients with fat mass accumulation, attributed to CKD-related wasting processes, reduced physical activity and possibly to obesity-induced inflammatory diseases, leading to sarcopenic obesity. Moreover, children with CKD are susceptible to abdominal obesity, resulting from high body fat distribution into the visceral abdomen compartment. Both sarcopenic and abdominal obesity are associated with increased cardiometabolic risk. This review analyzes the pathogenetic mechanisms, current trends and outcomes of malnutrition patterns in pediatric CKD. Moreover, it underlines the importance of body composition assessment for the nutritional evaluation and summarizes the advantages and limitations of the currently available techniques. Furthermore, it highlights the benefits of growth hormone therapy and physical activity on malnutrition management.

## 1. Introduction

According to the World Health Organization (WHO), malnutrition describes the imbalance between an individual’s energy and/or nutrient intake and energy requirements, and embodies three conditions: undernutrition, micronutrient-related malnutrition and micronutrient excess. In Western countries, although the prevalence of undernutrition is currently low, affecting less than 3% of the European population, pediatric overweight and obesity prevalence has dramatically increased from 4% in 1975 to 18% in 2016, affecting over 340 million children and adolescents aged 5–19 years old worldwide [1]. In the latest WHO European Region report, 29% of children aged 6–9 years old were overweight or obese with the highest prevalence observed in southern European countries [1]. Changes in eating behavior patterns, including the increased intake of fast food meals, sweet snacks, soft drinks and the reduced consumption of fruits and vegetables are well documented by the WHO European Childhood Obesity Surveillance Initiative [2], and primarily determine the risk of fat mass accumulation, leading to overweight and obesity. Moreover, daily physical activity level remains insufficient in the European adolescent population, mainly due to the increased time spent on screen activities and mobile devices. High adiposity levels coupled with sedentary behaviors, ultimately result to sarcopenic obesity, which varies from 5.66% to 69.7% in girls and from 7.2% to 81.3% in boys among different pediatric studies and is highly linked to the occurrence of non-communicable diseases, including cardiovascular disease and diabetes in early adulthood [3].

Children with CKD are susceptible to undernutrition and protein energy wasting (PEW) due to the CKD-related nutritional disturbances and inflammation-related wasting process (Table 1) [4]. Nevertheless, in parallel to the general pediatric population, the overweight and obesity prevalence in children with CKD has risen, attributed to growing adaptation of unhealthy eating habits and high exercise deficit disorder (Table 1) [5]. Both underweight and obesity undermine pediatric health, affecting renal function, cardiovascular risk and overall quality of life. Cumulative clinical data underline that skeletal muscle mass and strength deficits, which represent the principal finding of PEW and frailty, may also be present in children with high adiposity, due to CKD-related wasting processes, lack of physical activity and possibly due to obesity-induced inflammatory disease, increasing the risk of sarcopenic obesity (Table 1). Moreover, high adiposity and uremia-related pathogenetic mechanisms may also enhance body fat redistribution from the subcutaneous to visceral abdomen compartment, resulting in abdominal obesity (Table 1). Growing pediatric clinical studies emphasize that both high adiposity and abdominal obesity are deleterious for the cardiometabolic profile of these patients. This review describes the pathogenetic mechanisms, the current trends and outcomes of malnutrition patterns in children with CKD, according to recent clinical studies. Moreover, it analyzes the variety of malnutrition patterns according to body composition indices and outlines the advantages and the limitations of the currently available body composition techniques. Furthermore, it highlights the importance of integrating growth hormone therapy and physical activity in the malnutrition management of pediatric patients.

## 2. Malnutrition Patterns and Outcomes in Children with CKD

Undernutrition is a form of malnutrition, where nutrient intake is insufficient to meet the individual’s energy requirements, resulting in underweight, which is defined as body mass index (BMI), calculated with the following equation: BMI = weight (kg)/height (m)^2^, <5th percentile for age and sex (Table 1). Pediatric chronic undernutrition affects both weight and height growth, ultimately leading to poor height velocity and short stature; the latter is defined as height < 3rd percentile for age and sex [12]. The term undernutrition was initially used to describe starvation, where weight loss is mainly attributed to fat store depletion [13,14]. Protein energy malnutrition (PEM) was afterwards used for the cases where nutrient deficiency lead to both increased protein and fat catabolism [13,14]. Protein energy wasting (PEW) is used to describe the state of decreased body stores of protein (body muscle) and energy (body fat) fuels in CKD patients and reflect the nutritional disturbances as well as the CKD-related wasting processes (Table 1) [7]. The proposed diagnostic criteria for PEW by Abraham AG et al. in pediatric populations include reduced or loss of body mass and muscle mass, poor height growth and reduced caloric intake or decreased appetite, combined with abnormal biochemical parameters levels, including serum albumin, cholesterol and transferrin (Table 1) [8,9]. Nevertheless, definite diagnostic criteria are currently lacking. On the other hand, overnutrition is a form of malnutrition due to excessive nutrient intake, which exceeds the individual’s energy requirements. This condition induces abnormal body fat accumulation and ultimately results in increased BMI, clinically presented as overweight and obesity, defined as BMI > 85th percentile to <95th percentile and BMI> 95th percentile for age and sex, respectively.

Both undernutrition and overnutrition are linked to multiple comorbidities in pediatric CKD population (Figure 1). Weight loss and obesity were associated with faster progression to end stage CKD [15,16,17]. Underweight and obesity were associated with higher mortality rates in children on dialysis [18,19,20,21] and in kidney transplant recipients [22,23,24], forming a U-curve between BMI and mortality rate. Additionally, obese kidney transplanted patients may present a higher risk of allograft rejection dysfunction, allograft failure [23,25] and limited access to kidney transplantation from living donors [26]. Moreover, overweight and obese patients are at higher risk of hypertension, left ventricular hypertrophy and arterial stiffness, deteriorating cardiovascular outcomes [27,28] and are prone to metabolic disorders, including insulin resistance, hyperuricemia and disturbed lipid profile, augmenting the incidence of metabolic syndrome [29,30]. Nevertheless, the findings are inconsistent regarding the connection of undernutrition with cardiovascular risk and the occurrence of malnutrition–inflammation atherosclerosis syndrome in pediatric patients [28,31]. Although a U-curve correlation was observed between BMI z-score and pulse wave velocity in a single center study, no association was observed between malnutrition status and carotid intima media thickness in another study on chronic dialysis children [28,31]. Furthermore, limited adult and pediatric studies suggest that overweight patients with moderate CKD may present secondary hyperparathyroidism earlier in the course of the disease, indicating a link between obesity and mineral bone disorders [32,33,34]. In the pediatric population, this condition may be more frequent in kidney transplant recipients, where obesity prevalence is higher, as observed in a recent study [35].

## 3. Trends in Malnutrition Patterns and Risk Factors

The prevalence of underweight is currently low, affecting 8.9% of patients on peritoneal dialysis (PD) according to the International Pediatric Peritoneal Dialysis Network (IPPDN) database [21], 8.7% of kidney transplant recipients according to the USA Organ Procurement and Transplantation Network (OPTN) [23], and only 3.5% of patients on renal replacement therapy according to the European Society for Pediatric Nephrology/European Renal Association-European Dialysis and Transplant Association Registry (ESPD/ERA-EDTA) [5]. Nevertheless, according to the “CKiD” database, which constitutes a multicenter cohort registry of children with mild-to-moderate CKD in North America, the prevalence of PEW varies from 7 to 20%, depending on the applied criteria [9]. The discrepancy between underweight and PEW prevalence may actually indicate that current strategies for the management of the nutritional problems attributed to poor appetite, vomiting, overhydration, loss of nutrients during dialysis, dietary restrictions, etc., are relatively satisfactory but inflammation-related wasting processes remain insufficiently encountered [4]. 

Moreover, despite improvement in terms of weight growth, short stature remains high and varies from 7 to 44% in European countries, confirming that nutrition may only in part enhance height growth in children with CKD [36]. In addition, excessive nutritional supplementation disproportion to height gain during infancy and early childhood might ultimately lead to short and overweight children [37,38]. Sienna JL et al. observed that gastrostomy feeding significantly increased weight and BMI but not height in 20 CKD stage 2–5 children [39]. These findings were further confirmed by the last IPPDN study, where gastrostomy feeding was more frequent in overweight and obese PD patients and led to further BMI rise [21]. Monitoring of energy intake to support growth is therefore important in order to avoid overnutrition situations. 

On the other hand, the prevalence of both overweight and obesity in the CKD pediatric population has dramatically risen recently. According to the 2000 National (USA) Center for Health Statistics database, the BMI z-score of CKD pediatric patients referred to a tertiary service significantly increased from a median of +0.20 to +0.32 in two 8.5-year study periods [40]. Overweight and obesity actually affects 20.8% and 12.5% of patients on renal replacement therapy, according to ESPN/ERA-EDTA database [5], 10.7% and 22% of patients according to recent “CKiD” data [16] and 19.7% of patients at the start of PD according to IPPDN data [21].

Multiple factors may explain this trend (Figure 1). Dietary pattern changes play a significant role on the globally rising occurrence of obesity in the general pediatric population [1]. High-fat Western diets, irregular meal patterns, reduced vegetable consumption and inadequate fiber intake determine the risk of pediatric obesity. Energy intake seems to affect fat mass in CKD patients [41] and current literature data suggest a tendency toward an imbalanced diet lifestyle [42]. Although in earlier studies caloric intake appeared reduced in CKD patients [43,44], recent data from the “CKiD” registry indicated that caloric consumption surpassed the recommended intake in all age groups and was comparable to that of healthy populations [45]. In addition, energy intake was mostly derived from fast food products, while fruit and vegetable consumption was limited [46]. Interestingly, low dietary fiber was also reported by El Amouri A et al., especially in the advanced stages [47]. Apart from the global tendency toward a high-fat Western diet, alterations of smell and taste, observed in pediatric CKD patients [48], might also induce the patient’s preference to saltier foods high in saturated fats.

Moreover, according to a “CKiD” study, physical activity was lower and screen time exposure higher in CKD adolescents, compared to healthy controls, and especially in those with obesity and lower GFRs [49]. Physical performance was also found to be reduced in both pediatric transplant and dialysis patients [50,51]. The pathogenesis of poor exercise tolerance in pediatric CKD is not fully understood [52]. Anemia and cardiac changes have been implicated in its pathogenesis [52,53]. Reduced physical literacy, defined as lack of confidence and motivation in physical activity due to psychological factors, was observed in the CKD pediatric population [52]. Furthermore, sleep/rest fatigue, mostly observed in dialysis and kidney transplant recipients, was also associated with poor physical functioning [54]. Moreover, dynapenia, which will be described in the following sections may also contribute to impaired physical performance. 

## 4. Muscle Wasting and Outcomes in Children with CKD

Muscle wasting generally describes the muscle mass and strength wasting process observed in various chronic diseases, while sarcopenia mostly refers to the loss of muscle mass and strength (dynapenia) that occurs with aging and is highly prevalent in elderly individuals [55]. In the CKD pediatric population, both notions may be used to determine the disease-related muscle deficits. Muscle wasting is most frequently observed in large skeletal muscles, and increases with chronicity and progression of the disease, affecting more than 40% of patients on chronic dialysis [8,56,57]. According to the European Working Group on Sarcopenia in Older People (EWGSOP), sarcopenia is defined as the presence of low mass strength and muscle quantity or quality; severe sarcopenia is when physical performance is compromised; and reduced muscle strength is the principal diagnostic parameter [6,58] (Table 1). Nevertheless, there is a lack of established sarcopenia definitions and recommended techniques for muscle mass and function measurements in pediatric populations [59]. Moreover, we currently do not know whether dynapenia concurs with muscle mass loss in children with CKD. Muscle strength was significantly correlated to muscle mass in a recent pediatric study [60], while muscle force relative to muscle size was reduced in a study evaluating muscle torque to calf muscle cross-sectional area [61]. 

The growing body of clinical evidence in CKD pediatric populations underline that muscle wasting is associated with general fatigue, deteriorating life quality, and with decreased exercise capacity, jeopardizing cardiorespiratory fitness [54,56,62,63]. Moreover, although the muscle–bone unit is impaired in CKD, muscle wasting possibly contributes to bone deficits, resulting in decline of musculoskeletal health [56,61,64,65]. In clinical studies, reduced muscle strength was correlated to low bone cortical area and section modulus, suggesting that muscle impairment may enhance CKD bone disease [61,65]. The separate unfavorable outcomes of sarcopenia and dynapenia in pediatric CKD have not been distinguished yet. In adult studies, dynapenia was associated with GFR decline, mortality and cerebrovascular events, independently of muscle mass loss, highlighting the distinct adverse effects of muscle dysfunction in CKD patients [66,67,68]. 

Progression of muscle wasting ultimately develops a frailty phenotype, characterized by an increased vulnerability to adverse health outcomes [10,64]. This clinical syndrome has been thoroughly investigated in adult CKD patients. Several diagnostic tools have been validated; the most common, called the Fried phenotype, includes the following criteria: weight loss, fatigue, weakness, slow walking speed and reduced physical activity (Table 1) [11]. The frailty phenotype has been recently described in children with CKD by Sgambat K et al. it was associated with a higher hospitalization risk, and the suggested criteria include suboptimal growth/weight, low muscle mass, fatigue and high C-reactive protein (CRP) levels (Table 1) [10]. Low circulating insulin growth factor 1 (IGF-1) levels was also identified as a risk factor [64]. Further data on the outcomes and biochemical biomarkers of frailty are needed, particularly in children on chronic dialysis, where the prevalence is considerably higher [64]. 

In clinical practice, muscle wasting may be observed in both undernutrition and overnutrition conditions (Figure 1). In the following, we will present the patterns of malnutrition which may concur with muscle wasting in CKD pediatric patients. 

## 5. Undernutrition/PEW and Muscle Wasting in Children with CKD

Muscle wasting is the cornerstone finding of PEW. According to “CKiD” data, Abraham AG et al. found that reduced mid-arm circumference (MAC) was more prevalent than reduced body mass in all CKD stages, affecting 41% and 25% of patients, respectively, suggesting that muscle wasting precedes underweight occurrence [9]. In the same study, overall poor growth prevalence was 42% and was approximately the same as that of reduced muscle mass in all CKD stages, proving that poor growth and muscle wasting mostly occur concurrently. 

Reduced protein intake, principally due to uremic anorexia, may only in part explain sarcopenia [56]. Cachexia is the term used in adult populations to define the muscle wasting process due to the complex metabolic syndrome associated with underlying illness with or without loss of fat mass [69]. In CKD, PEW pathogenic mechanisms, including systemic inflammation, triggered by the accumulation of uremic toxins, immune cell defects, gut dysbiosis and dialysis-related factors, metabolic acidosis, disrupted growth hormone (GH)–IGF-1 axis, insulin resistance, anemia, mineral bone disorders and activation of the renin–angiotensin system have been implicated in the muscle mass and strength wasting process (Figure 1) [56]. These multiple pathways impair the balance between muscle catabolic and anabolic processes, leading to degradation of muscle proteins, through activation of the ubiquitin–proteasome system (UPS), reduced muscle protein synthesis, inhibited myogenesis and raised muscular energy expenditure [56]. Current clinical data highlight that apart from muscle quantity, muscle quality is also degraded in CKD, involving muscle fibrosis (myofibrosis) and intramuscular fat infiltration (myosteatosis) [70]. Experimental and longitudinal clinical adult studies have shown that kidney dysfunction promotes ectopic fat redistribution and progressive fat accumulation in skeletal muscles, inducing structural changes [71,72]. Uremic toxin accumulation, gut dysbiosis, and adipokine imbalance are possible incriminating factors [72]. Myosteatosis contributed to decreased physical functioning and muscle strength in a cohort of pediatric patients on peritoneal dialysis [63] and was associated with poor cardiovascular outcome in adult patients [72]. 

Data regarding the prevalence of fat mass loss in CKD pediatric patients are limited. Bioimpedance spectroscopy (BIS)-based body composition assessment revealed that low lean tissue index was observed in 22.6% of patients with moderate CKD and in 36.7% of patients with advanced CKD, while fat tissue index was observed in only 10% of patients with advanced CKD in a cross-sectional study on 61 patients [33]. Iyengar et al. found body fat using Dual-energy X-ray absorptiometry (DXA) and bioimpedance analysis (BIA) was reduced in 18% and 12% of children with CKD 2-5D, respectively [73]. Canpolat et al. found that reduced mid-arm muscle circumference (MAMC) and decreased BIA-based fat mass were present in 60% and 21% of patients on chronic dialysis, respectively [31]. These findings support that muscle wasting largely precedes fat loss in the CKD pediatric population [4]. 

## 6. Obesity and Muscle Wasting

### 6.1. General Population

Current literature data indicate that BMI cannot distinguish lean from fat mass, reducing its specificity to detect body lean mass and adiposity [74]. There is a need for prompt diagnosis of fat mass accumulation and early management of associated health consequences to detect two conditions, where high body fat percentage is present in individuals with normal BMI, called normal-weight obesity (NWO), or with concurrent sarcopenia, called sarcopenic obesity (Table 1) [75]. High body fat percentage cut-off points, proposed by the WHO, are established as 25% and 30% for adult men and women, respectively (Table 1). Nevertheless, in pediatric populations, cut-off levels are not globally defined and vary between the 80 and 90 percentile for age and sex among different studies (Table 1). NWO is identified as a risk factor for adverse metabolic and cardiovascular outcomes in pediatric patients and was associated with impaired physical fitness, including reduced hand grip strength, exercise deficit disorder and decreased motor performance, ultimately leading to sarcopenic obesity [76,77,78,79]. Sarcopenic obesity in turn was also linked with impaired cardiometabolic and mental health [3]. 

Apart from the reduced physical activity frequently observed in patients with high adiposity, obesity per se may also contribute to the occurrence of sarcopenia. Adipose tissue is nowadays considered as an endocrine organ and obesity is considered a chronic low-grade inflammatory disease. There is increasing literature evidence that high adiposity levels induce the secretion of systemic pro-inflammatory cytokines, enhancing protein muscle catabolism, while impairing the GH/IGF-1 axis and inhibiting the repair and regeneration of skeletal muscles [80]. Obesity has been also associated with damaging skeletal muscle contractile function, by inducing a shift from slow to fast muscle fiber types [81]. Furthermore, the alteration of the abundance and diversity of the gut microbiota, induced by unhealthy dietary patterns, coupled with the reduced physical activity, may promote insulin resistance, further enhancing muscle wasting [82]. 

Individuals with a high body fat percentage are also prone to abdominal obesity, describing the state of increased visceral abdominal fat, which is strongly associated with adverse cardiometabolic profiles [83]. Although high central adiposity level has a strong genetic basis, positive energy balance is the primary cause of overflow of subcutaneous fat storage and redistribution of fat into intra-abdominal tissue [83]. Waist circumference (WC) was initially used to describe central adiposity levels in both adult and pediatric populations. Nevertheless, WC is a height-dependent parameter, and its application for measurement of central adiposity may lead to overestimation and underestimation in taller and shorter individuals, respectively [84]. Adjustment of WC to body size by calculating the waist-to-height ratio (WtHr), with the following equation: WtHr = waist (cm)/height (cm), seems to be a better screening tool for cardiometabolic risk assessment in adults and a cut-off of 0.5 is generally applied for both adult and pediatric populations [84]. Most pediatric studies suggest that WtHr predicts total body fat better than BMI [85,86], and equally to body composition devices, such as DXA [87]. Although literature data regarding the benefits of WtHr measurement in normal weight pediatric patients are inconsistent [88], WtHr is a strong cardiometabolic risk parameter among overweight and obese children [89,90].

### 6.2. Children with CKD

Growing clinical data indicate that children with CKD are susceptible to NWO, sarcopenic as well as abdominal obesity. According to a cross-sectional study on 41 patients with CKD 2-5D, NWO, defined as normal BMI and relative fat mass (RFM) > 85th percentile, was present in 17% of patients and 46% of those with high RFM [30]. Moreover, high WtHr was observed in 15% of CKD patients with normal BMI in a study from the “CKiD” registry and accounted for approximately 31% of patients with high adiposity, estimated by either BMI or WtHr [91]. Interestingly, Rashid et al. found a discordant relative high fat mass and low lean mass in a single-center cohort of 100 kidney transplant and CKD children based on DXA findings with a higher trunk-to-leg fat mass ratio [41]. Foster et al. found reduced DXA-based leg lean mass but increased body fat mass in 143 CKD patients. In further analysis, they observed that trunk but not leg fat mass was significantly higher in all CKD groups [92]. Muscle strength was also found to be impaired in overweight CKD pediatric patients. In kidney transplant recipients, dynapenia was more frequent in overweight patients [62,93], while according to the “CKiD” database, hand grip strength was reduced in children with CKD, regardless of BMI status [57]. In further analysis, Hogan et al. observed that among patients with reduced hand grip strength, 34% were overweight or obese and only 1% were underweight [57]. 

The reduced physical activity frequently observed in CKD pediatric patients, coupled with the CKD and possibly the obesity-induced muscle wasting process may explain these findings. Moreover, a uremic milieu per se may also enhance fat redistribution from the subcutaneous compartment to visceral compartment and principally in the abdomen [94]. Although the causal effects are not well understood, Aguilera et al. suggest that the uremia-induced hyperinsulinemia, disorders of orexigenic and anorexigenic hormones, adipokine imbalance, reduced IGF-1 bioavailability and secondary hyperparathyroidism, which are implicated in muscle wasting and therefore the PEW process, may promote dysregulation of fat distribution and abdominal fat accumulation without an increase in appetite [94]. Further studies are required to elucidate the pathogenetic mechanisms of abdominal obesity in CKD patients. Therefore, a wide range of malnutrition patterns may be observed in children with CKD, based on body composition indices, varying from non-obese sarcopenia to obesity and all malnutrition types may be associated with abdominal obesity (Figure 2).

Cumulative clinical studies support the clinical significance of high total or abdominal adiposity in CKD pediatric patients (Table 2). According to analysis of the “CKiD” registry, WtHr was more sensitive than BMI in predicting cardiovascular risk in both CKD [91] and kidney transplant patients [95]. Moreover, normal weight patients with high BIA-based fat free tissue/fat tissue index [FFTI/FTI] ratios presented lower grades of arterial stiffness [28], while those with BIS-based high RFM were susceptible to insulin resistance and hyperuricemia in two cross-sectional studies [30]. Furthermore, DXA-based fat mass index was significantly associated with hypertension, after adjusting for the presence of obesity in a cohort of 63 CKD pediatric patients [96]. In addition, children with cardiometabolic risk parameters presented a faster decline in kidney function, regardless of BMI status [97]. Nevertheless, a longitudinal study in the “CKiD” registry found that waist circumference (WC) added limited information to BMI for the prediction of metabolic, cardiovascular and renal outcomes [29]. Further large-scale studies are needed to investigate the prevalence and the cardiometabolic and renal outcomes of high adiposity in CKD pediatric patients with normal BMI and/or concurrent muscle wasting. 

## 7. Body Composition Assessment

The suggested approach for nutritional evaluation mostly includes the assessment of protein and energy intake and the measurement of the traditional anthropometric parameters. Caloric intake is usually calculated based on a 3-day diet record. Estimation of appetite based on a 5-point severity scale could also serve as a useful marker of PEW [98]. All anthropometric parameters values, including weight, height, head circumference, height velocity, BMI or weight to length, should be converted to z-scores based on percentile reference growth chart curves according to the clinical practice recommendations from the Pediatric Renal Nutrition Taskforce [98]. Moreover, adjustment of BMI to height–age is essential in patients with poor height growth, for a better evaluation of relative lean mass [98]. 

Although body composition assessment is required for proper evaluation of malnutrition in children with CKD, there are currently no validated methods. Anthropometric techniques are inexpensive, simple to perform and noninvasive but their application in clinical practice involves many limitations:WtHr seems to be a promising marker of abdominal adiposity and related cardiovascular risk in large-scale CKD pediatric studies. However, the cut-off for high WtHr in children with CKD is not defined. Moreover, abdominal adipose tissue consists of both subcutaneous and visceral fat deposits. In adult CKD patients, visceral fat was strongly correlated to WtHr in CKD patients [99], but similar studies are lacking in pediatric populations [86]. In addition, WtHr may overestimate body fat in children with clinical edema and may be unsuitable in children on peritoneal dialysis [100].Skinfold thickness (SFT), used for the measurement of subcutaneous fat mass, is operator-dependent and generates biased estimates of body fat compared to DXA in the general pediatric population [101]. Moreover, this parameter does not provide information on visceral fat [99].Mid-arm circumference (MAC) and mid-arm muscle circumference (MAMC) are applied for the measurement of mid-arm muscle mass. Specifically, MAMC is calculated based on SFT and MAC levels, based on the following equation: MAMC = MAC − (3.1415 × triceps SFT). In adult CKD studies, higher MAMC and MAC were significantly associated with better survival rates in patients on chronic hemodialysis [102,103]. In pediatric patients, MAC was suggested as a marker of PEW [8,9]. Nevertheless, these anthropometric techniques are currently not recommended by KDOQI guidelines for routine nutritional evaluation of pediatric CKD patients [104]. Firstly, their levels vary according to patient hydration status and the high intra-observer and inter-observer variability limits their accuracy. In addition, the lack of MAC to differentiate between mid-arm subcutaneous fat and muscle mass, further reduces its value for estimation of the actual patient muscle mass status. In the general pediatric population, MAC was more useful for predicting regional fat rather than muscle mass [105]. Moreover, possible abnormal muscle regional distribution limits their precision for estimation of total body muscle mass [106].

Several tools have been applied for body composition assessment in general pediatric populations and in children with CKD. Due to the large discrepancy of body compartment results among the different methods, it is necessary that the same method is applied for patient nutritional monitoring in order to acquire comparable results (Table 3) [73,107,108,109,110,111].

Each method embodies limitations and some extent of measurement error [100,107]. We will present the pros and cons of the most applied methods in children with CKD (Table 4):DXA is quick method, based on a three-compartment model, and estimates the amount of total body fat, bone mineral and bone-free fat-free mass, according to tissue density properties [59,107]. It is widely considered as a precise method for body composition measurement in the general pediatric population [59,107] and it is validated by the EWGSOP for muscle quantity evaluation [6]. Another advantage of DXA is that it provides segmental body composition assessment, including the measurement of appendicular skeletal muscle and trunk-to-leg fat mass ratio. Nevertheless, the small dose of ionizing radiation limits its application in clinical practice for regular body composition evaluation. Furthermore, possible altered tissue density and hydration status in CKD patients may decrease the accuracy of its results [59,100,106].Air-displacement plethysmography (ADP) is a quick, non-invasive method, based on a two-compartment model, and uses tissue volume to calculate tissue density properties and estimate fat and fat-free mass [59,107]. It is equal to DXA in terms of accuracy for body composition assessment [59]. The disadvantage of this technique is that is expensive and not easily available in clinical practice and, as with DXA, it is less suitable for individuals with excess fluid retention and under-mineralization [59,100,106].Bioimpedance analysis (BIA) is a convenient, easy to perform, bedside tool, which indirectly calculates a two-compartment model, including fat mass and fat-free mass, according to tissue electrical impedance (resistance and reactance) properties. Although its accuracy on an individual level is reduced compared to DXA and ADP [59], it is approved by the EWGSOP for determination of muscle quantity [6]. Moreover, its affordability, portability and relatively low cost make it the most appealing method for nutritional evaluation monitoring [6]. Nevertheless, since total body water (TBW) measurement is used to estimate lean body mass, patient overhydration may lead to overestimation of fat-free mass and underestimation of fat mass [106]. Moreover, BIA-derived results are influenced by the total level of body fat mass, given that the extracellular water (ECW)-to-TBW ratio is greater in obese individuals and lower in lean individuals [59,107]. The multifrequency BIS distinguishes TBW into intracellular water (ICW) and ECW compartments and provides a three-compartment model, including lean mass, adipose tissue mass and overhydration compartment [107]. Moreover, it estimates body cell mass (BCM), consisting of lean and intracellular lean water masses [107]. This method has been recently validated for hydration status assessment and provides an excellent intra- and inter-rater reproducibility in the pediatric population [111], while it seems to be a valuable prognostic tool, at least at a population level, in CKD adult patients [112]. Further studies are needed to determine its utility in CKD pediatric populations.Whole body potassium scanning (TBK) measures BCM, since total body potassium is proportional to BCM, and subsequently provides an estimation of lean mass [106]. Nevertheless, this technique is not widely available and the increased tissue K concentration possibly observed in CKD limits its accuracy [106].Isotope dilution measures TBW and subsequently estimates fat-free mass. Although this technique is easily applied in pediatric patients, it is not accessible in most centers and hydration variability may skew its results in CKD pediatric patients [106].

## 8. Assessment of Muscle Strength and Physical Performance

Handgrip strength is a good marker of total muscle strength in healthy children over the age of 6 years old and adolescents [113] and is approved by the EWGSOP as a simple, objective, inexpensive and reliable tool for muscle strength assessment in adults [6]. In the general pediatric population, reduced handgrip strength was associated with dysregulated glucose metabolism [114] and hypertension [115], while adequate grip strength attenuated the adverse effects of obesity on cardiometabolic risk [116]. Moreover, adjustment of handgrip strength to BMI was recently proposed as a marker of sarcopenic obesity [117]. Further studies are required to define the adverse effects of dynapenia in CKD pediatric patients and the optimal method for adjustment of hand grip strength levels to concurrent poor height growth. Moreover, additional data are needed to the evaluate the correlation between upper and lower extremities strength in children with CKD. 

Validated age-appropriate methodological approaches for evaluation of overall physical performance are not available in pediatric populations [59]. Physical activity questionnaires for children (PAQ-C) and adolescents (PAQ-A) have been proven effective for evaluating exercise deficit disorders in CKD pediatric patients [50,51]. Estimation of fatigue, adapted to pediatric population multidimensional fatigue questionnaires (PedsQL-MFS) may be also useful for the diagnosis of frailty and poor physical functioning [54,64]. Reduced exercise capacity in adult populations is mostly assessed by using the Short Physical Performance Battery (SPPB), which includes usual gait speed, 6 min walk test (6MWT) and the stair climb power test [6]. Among those tests, 6MWT performed in a 20 m-long track in a straight hallway was used for the evaluation of physical status in pediatric patients [118] and may be beneficial for the monitoring of physical performance. Gross motor skills may be also impaired in children with CKD [119]. Nevertheless, data are lacking regarding the motor performance and the need for motor skill assessment in infant and toddler patients. 

## 9. Recombinant Growth Hormone Therapy

Recombinant growth hormone (rGH) therapy has been proven efficient for promoting height growth in children with CKD and is currently recommended in children aged above 6 months with CKD 3D who present short stature and low height velocity, the latter defined as below the 25th percentile for age and sex [120]. Apart from the benefits on height growth, limited pediatric CKD longitudinal studies advocated that rGH therapy may also enhance body muscle mass and slightly decrease body fat mass (Table 5) [56,121,122,123,124]. The promoting effects of GH on muscle growth are mainly mediated by liver- and skeletal muscle-secreted IGF-1, which stimulates amino acid transport, enhancing protein synthesis and reducing protein breakdown and oxidation [125,126]. In parallel, GH may induce lipolysis, by inhibiting the lipid storage action of lipoprotein lipase and stimulating hormone-sensitive lipase action, which breaks down adipose tissue-stored triglycerides [127]. According to a cohort of 594 pediatric patients from 12 European countries, rGH therapy was administered in only 15% of children with growth retardation, while lower height was more frequently observed in overweight patients [36]. We suggest that application of rGH since infancy in parallel with optimal management of nutrient intake may prevent the double burden of malnutrition, including stunting and wasting, and in parallel reduce the risk of fat mass accumulation later in life. Further studies are required to confirm whether early onset of rGH therapy may prevent the risk of overweight and short stature in CKD pediatric patients. 

## 10. Physical Activity

Physical activity is highly encouraged in adult CKD patients with sarcopenia; resistance exercise promotes muscle mass and strength, while endurance training enhances physical performance and exercise capacity [55]. Moreover, interdialytic training has been successfully administered in various adult CKD 5D cohorts and seems beneficial in promoting muscle anabolic processes [55,128]. Similar exercise training programs have been applied in a few pediatric studies with no reported adverse events or side effects [129]. Nevertheless, the benefits in terms of functional fitness and muscle strength improvement were limited, mainly due to high drop-out rates, attributed to the lack of patient motivation and the short program duration [52]. Moreover, most programs included only aerobic exercise, probably because resistance exercises may further discourage the children from participating in physical activities [129]. A multidisciplinary approach, including the presence of a qualified sport scientist, who will design and adapt the training program according to the patient needs, and possibly of a qualified psychologist, who will support and encourage the child in undertaking physical exercise, is probably the key for successful implementation of training programs in CKD pediatric patients [129]. Moreover, management of possible sleeping disorders, encouragement of child to limit the amount of recreational screen time and participate in school activities may also further enhance the patient’s physical performance.

## 11. Conclusions

Children with CKD are susceptible to both undernutrition/PEW and to overnutrition/obesity patterns of malnutrition. Both conditions may ultimately imperil renal function and cardiovascular health. Τhe risk for high adiposity in patients with normal BMI and the growing prevalence of muscle wasting in both PEW and obese conditions, indicate the need for body composition assessment. Application of anthropometric techniques are, in general, discouraged due to the lack of precision and the high inter- and intra-observer variability. Several technological tools are used in clinical practice and regardless of their limitations, they could be proven efficient for body composition monitoring. Further studies are needed to evaluate the benefits of regular body composition monitoring in everyday clinical practice. Moreover, apart from nutritional management, early-onset rGH therapy in case of growth failure and application of exercise training tailored to the patient needs may actually help to improve muscle mass and strength and in parallel reduce the risk of fat mass accumulation.

## Figures and Tables

**Figure 1 life-13-00713-f001:**
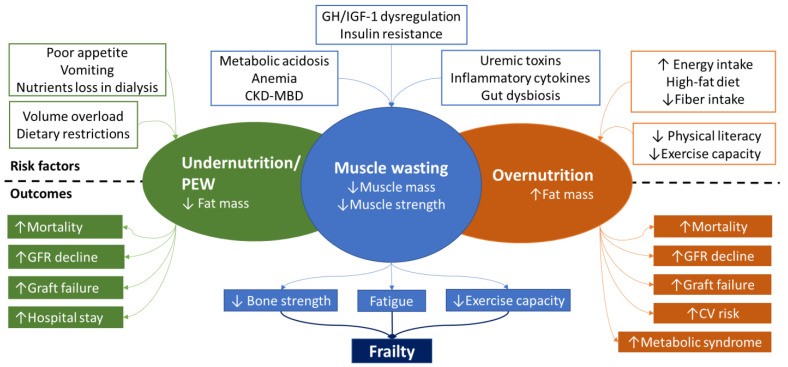
Risk factors and outcomes of malnutrition in children with chronic kidney disease. (Abbreviations: CKD-MBD: chronic kidney disease-mineral and bone disorders, CV: cardiovascular, GFR: glomerular filtration rate, GH: growth hormone, IGF-1: insulin growth factor 1, PEW: protein energy wasting).

**Figure 2 life-13-00713-f002:**
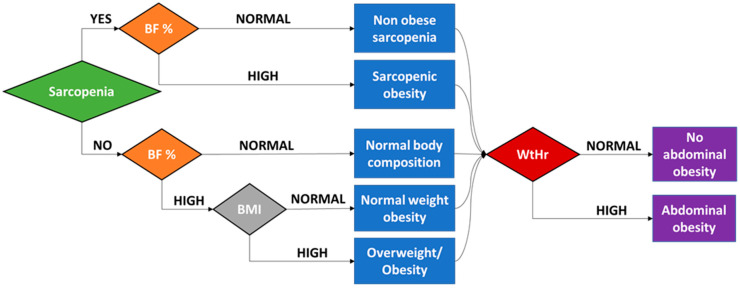
Malnutrition patterns based on body composition indices, including muscle mass, body fat percentage, body mass index and waist-to-height ratio, in children with chronic kidney disease. (Abbreviations: BF%: body fat percentage, BMI: body mass index, NWO: normal-weight obesity, WtHr: waist-to-height ratio).

**Table 1 life-13-00713-t001:** Definitions and suggested criteria of malnutrition terms in pediatric and adult populations with chronic kidney disease.

	Pediatric Population	Adult Population
Short stature	Height < 3rd perc for age and sex	
Growth failure	Height < 3rd perc for age and sex Height velocity < 25th perc for age and sex	
Underweight	BMI * < 5th perc for age and sex	BMI < 18.5
Overweight	85th perc < BMI * ≤ 95th perc for age and sex	25 ≤ BMI < 30
Obesity	BMI * > 95th perc for age and sex	BMI ≥ 30
High body adiposity	BF% > 80–90th perc for age and sex	BF% > 25% in men and >30% in women
Sarcopenia [6]	Muscle mass * < 5th perc for age and sex Severe sarcopenia: low physical performance	Low muscle strength and low muscle quantity or quality Severe sarcopenia: low physical performance
Normal weight obesity	High body adiposity and BMI * ≤ 85th perc for age and sex	High body adiposity and BMI < 25
Sarcopenic obesity	High body adiposity or overweight/obesity and sarcopenia	High body adiposity or overweight/obesity and sarcopenia
Abdominal obesity	WtHr > 0.5	WtHr > 0.5
Protein energy wasting [7,8,9]	Anthropometric parameters underweight or decrease in BMI * perc of ≥10% over 1 year sarcopenia or decrease in muscle mass * perc of ≥10% over 1 year short stature or decrease in height perc of ≥10% over 1 year Reduced appetite Abnormal biochemical parameters levels (serum albumin, cholesterol and transferrin)	Anthropometric parameters BMI < 23 or unintentional weight loss (5% over 3 months or 10% over 6 months) or BF% < 10% Decrease in muscle mass (5% over 3 months or 10% over 6 months) or low MAMC
		Low dietary intake Abnormal biochemical parameters levels (serum albumin, prealbumin, cholesterol)
Frailty phenotype [10,11]	Anthropometric parameters underweight or decrease in BMI * perc of ≥10% over 1 year sarcopenia or decrease in muscle mass * perc of ≥10% over 1 year short stature or decrease in height perc of ≥10% over 1 year Fatigue High CRP level	Criteria: weight loss low hand-grip strength decreased energy level impaired walking speed physical inactivity

BF%: body fat percentage, BMI: body mass index, CRP: C-reactive protein, MAMC: mid-arm muscle circumference, WtHr: waist-to-height ratio, perc: percentile. * BMI and muscle mass should be adjusted to height-age in case of short stature.

**Table 2 life-13-00713-t002:** Association between high adiposity levels and adverse cardiometabolic outcomes in children with CKD.

Author	No. of Patients	Technique	Results	Conclusions
Sgambat K et al. [91]	593 CKD patients	WtHr	Overweight patients based on both WtHr and BMI presented higher SBP z-scores (*p* < 0.001), LVMI (*p* < 0.001), glucose (*p* = 0.003), and triglyceride (*p* < 0.001) levels and lower HDL-C (*p* < 0.001) levels compared to lean patients Cardiovascular markers levels did not significantly differ between overweight patients and lean patients based on BMI only	WtHr should be used for cardiovascular risk screening in children with CKD
Sgambat K et al. [95]	42 kidney transplant recipients	WtHr	Area under the ROC curve to detect cardiovascular outcomes was higher for WtHr-obese (0.77) compared to BMI-obese (0.47) and WC-obese (0.48) Adverse cardiovascular outcomes were associated with WtHr obesity (*p* = 0.0001) but not with BMI or WC obesity	WtHr is more sensitive than BMI or WC to detect cardiovascular risk in pediatric kidney transplant recipients
Karava V et al. [30]	41 CKD patients	RFM	RFM, which corresponds to BF%, was a stronger predictor of high HOMA-IR compared to BMI height–age z-score in both total (AUC = 0.785 and AUC = 0.709, respectively) and normal-weight (AUC = 0.783 and AUC = 0.686, respectively) patients RFM was associated with high HOMA-IR in both total (OR 1.168, 95% CI 1.022–1.334, *p* = 0.022) and normal-weight (OR 1.285, 95% CI 1.021–1.619, *p* = 0.033) patients	Children with high RFM, including those with NWO, are at risk for insulin resistance
Karava V et al. [28]	26 CKD patients	FFTI/FTI	Among normal weight patients, FFTI/FTI ratio, calculated with the following equation: fat free mass (kg)/height (m)^2^/fat mass (kg)/height(m)^2^, ≥2.5 was significantly associated with lower PWV z-scores (*p* = 0.013)	Targeting FFTI/FTI ≥ 2.5 could be protective against cardiovascular disease in normal weight children
Hsu CN et al. [96]	63 CKD patients	FMI	FMI z-score, calculated with the following equation: fat mass (kg)/height (m)^2^, was independently associated with abnormal ABPM in multivariate logistic regression analysis (*p* = 0.037)	FMI may help discriminating cardiovascular risk in children and adolescents CKD
Patel HP et al. [29]	737 CKD patients	WC	BMI was more strongly associated with LVMI compared to WC (8.54%, 95% CI 6.47–10.65 vs. 5.79%, 95% CI 3.33–8.32) in the nonglomerular CKD group but both had significant associations	WC added limited information to BMI in terms of cardiometabolic risk prediction

ABPM: ambulatory blood pressure monitoring, BMI: body mass index, CKD: chronic kidney disease, FFTI/FTI: fat-free tissue index/fat tissue index, FMI: fat mass index, HDL-C: high density lipoprotein–cholesterol, HOMA-IR: homeostatic model assessment for insulin resistance, LVMI: left ventricular mass index, PWV: pulse wave velocity, RFM: relative fat mass, SBP: systolic blood pressure, WC: waist circumference, WtHr: waist-to-height ratio.

**Table 3 life-13-00713-t003:** Comparison of body composition methods in children with chronic kidney disease.

Author	No. of Patients	Techniques	Parameter	Results	Conclusions
Milani GP et al. [108]	16 CKD 5D	BIS vs. Deuterium and Bromide dilution	Total Body Water, Extracellular Water	TBW Bland–Altman analysis: mean difference between the two methods −0.09 L (95% CI −2.1–1.9) ECW Bland–Altman analysis: mean difference between the two methods +0.6 L (95% CI −2.3–3.5)	BIS measurements were imprecise compared to dilution techniques
Milani GP et al. [109]	15 CKD 5D	BIS vs. Deuterium dilution	Body Lean Mass	Bland–Altman analysis: mean difference (+/−SD) between the two methods 0.25 ± 2.30 kg (95% CI −4.80–4.25)	Wide limits of agreement between the two techniques
Wong Vega M et al. [110]	15 CKD 5D	BIA vs. ADP	Body Fat Mass	BIA-derived BFM was lower in males with Tanner ≥4 (*p* = 0.0004) and in obese subjects (*p* = 0.005) and higher in malnourished patients (*p* = 0.02) compared to ADP	BIA underestimated BFM in males with Tanner ≥4 and in obese patients and overestimated FM in malnourished patients
Iyengar A et al. [73]	33 CKD 2-5D	BIA vs. DXA	Body Fat Mass	FMI: good degree of reliability in CKD 2–5 (ICC 0.76 CI [0.48,0.9]), poor reliability in CKD 5D (ICC 0.58 CI [0.1,0.84]) BF%: poor to fair reliability in CKD 2–5 (ICC 0.64 [0.28,0.84]) and CKD 5D (ICC 0.53 [0.02,0.82])	BIA-derived FMI did not compare well with DXA-derived FMI in CKD 5D patients while FMI was comparable with a lower bias
Dasgupta I et al. [111]	6 CKD 5D 45 measurements	BIS vs. UKM	Total Body Water	Good TBW_BIS and TBW_UKM agreement (mean difference − 0.4 L, 2SD = ± 3.0 L)	BCM can be used in children with CKD to assess fluid status

ADP: air displacement plethysmography, BCM: body cell mass, BIA: bioimpedance analysis, BIS: bioimpedance spectroscopy, BF%: body fat percentage, BFM: body fat mass, CKD 5D: chronic kidney disease stage 5 on chronic dialysis, DXA: dual-energy X-ray absorptiometry, ECW: extracellular water, FMI: fat mass index, ICC: intra-class correlation, TBW: total body water, UKM: urea kinetic modeling.

**Table 4 life-13-00713-t004:** Advantages and limitations of body composition techniques.

Anthropometric parameters pros: simple, quick, non-invasive cons: high intra-/inter-observer variability	WtHr	pros: marker of abdominal adiposity cons: unsuitable in children with clinical edema or on PD
	Skinfold thickness	pros: marker of subcutaneous adiposity cons: cannot measure visceral adiposity, unsuitable in children with clinical edema
	MAC	pros: marker of muscle and fat mass, marker of PEW cons: limited precision for body muscle mass measurements, unsuitable in children with clinical edema
	MAMC	pros: marker of muscle mass, marker of PEW cons: unsuitable in children with clinical edema
Technological tools pros: high intra- and inter-rater reproducibility pros: not easily available in clinical practice	DXA	pros: quick, precise, measurement of body bone, fat-free and fat mass, measurement of segmental body composition (appendicular skeletal muscle, trunk/leg fat ratio) cons: ionizing radiation, high dependency on hydration and tissue density status
	ADP	pros: quick, precise, measurement of body fat-free and fat mass cons: expensive, high dependency on hydration and tissue density status
	BIA/BIS	pros: bedside tool, measurement of body fat and fat-free mass (BIA) or adipose, lean tissue mass and overhydration (BIS), measurement of BCM (BIS) cons: imprecise compared to DXA and ADP, high dependency on hydration status
	TBK	pros: measurement of BCM cons: imprecise for lean body mass estimation, unsuitable in cases of increased tissue K concentrations
	Isotope dilution	pros: measurement of TBW cons: imprecise for lean body mass estimation, high dependency on hydration status

ADP: air-displacement plethysmography, BCM: body cell mass, BIA: bioimpedance analysis, BIS: bioimpedance spectroscopy, DXA: dual-energy X-ray absorptiometry, MAC: mid-arm circumference, MAMC: mid-arm muscle circumference, PD: peritoneal dialysis, PEW: protein energy wasting, TBK: whole-body potassium scanning, TBW: total body water, WtHr: waist-to-height ratio.

**Table 5 life-13-00713-t005:** Body composition changes in children with chronic kidney disease under growth hormone therapy.

	No. of Patients	Duration	Methods	Results	Conclusion
Johnson VL et al. [121]	14	6 months	DXA	Increase in LBM (kg): 17.9+/−3.0 vs. 20.7+/−3.6, *p* = 0.04 Decrease in FBM (kg): 4.4 +/−1.4 vs. 3.−1.2), *p* = 0.002 Decrease in BF (%): 18.6+/−3.9 vs. 14.5+/−3.4, *p* = 0.04	rGH therapy enhances repletion of LBM
van der Sluis IM [122]	18 vs. 15 controls	24 years	DXA	rGH therapy patient group: Increase in LBM SDS: −1.96+/−0.86 vs. −0.86+/−1.29, *p* < 0.05 Decrease in BF% SDS: 0.00+/−1.07 vs. −0.96+/−0.90, *p* < 0.05 Non rGH therapy patient group: Non-significant change in LBM SDS: −1.17+/−0.62 vs. −1.24+/−0.78 Decrease in BF% SDS: −0.07+/−1.18 vs. −0.53+/−1.06, *p* < 0.05 Compared to controls, patients given rGH therapy presented higher LBM (*p* < 0.001) and lower BF% (*p* = 0.05)	LBM increased only in the rGH therapy patient group
Boot AM et al. [123]	17 vs. 19 controls	12 months	DXA	rGH therapy patient group: Increase in LBM SDS: −1.98+/−0.88 vs. −1.09+/−1.08, *p* < 0.001 Decrease in BF% SDS: −0.05+/−1.08 vs. −0.95+/−0.94, *p* < 0.001 Non-rGH therapy patient group: Non-significant change in LBM SDS: −1.19+/−0.63 vs. −1.27+/−0.62 Non-significant change in BF% SDS: −0.09+/−1.15 vs. −0·54+/−1.17	LBM increased and BF% decreased only in the rGH therapy patient group
Vaisman N et al. [124]	8	12 months	DXA TBK MAMC	FBM (kg): 2.9 +−1.1 vs. 2.8+/−0.9, *p* = NSTBK (mEq): 1099+/−311 vs. 1078+/−220, *p* = NS MAMC (cm): 158.7+/−19.4 vs. 165.9 +/−23.3, *p* = 0.01	rGH therapy had a positive effect on MAMC, but not on LBM (as reflected by TBK) and FBM

BF%: body fat percentage, DXA: dual-energy X-ray absorptiometry, FFM: fat-free mass, FBM: fat body mass, LBM: lean body mass, MAMC: mid-arm muscle circumference, rGH: recombinant growth hormone, TBK: whole-body potassium scanning.

## Data Availability

Not applicable.

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
