# Peer review of "Malnutrition Patterns in Children with Chronic Kidney Disease"

_life, 2023, doi:10.3390/life13030713_

Round 1

Reviewer 1 Report

In the present review article, authors closed-up susceptible undernutrition and protein energy wasting related to chronic kidney disease (CKD) and focused obesity and the accompanied/potential sarcopenia in pediatristic patients with CKD. This review describes the pathogenetic mechanisms, the current trends and outcomes of malnutrition patterns in children with CKD, according to recent clinical studies. I thought that their focus was interesting and was mostly well summarized in the text. I listed some issues to improve the manuscript as below.

Major comments

1.       It may remain unsure what different problems will be between adults and children with CKD. Although I was misreading, I guessed that some session might involve involves mixed with children and adults’ status about assessment of nutritional descriptions.     It may be important to show separately and ideally show tables to compare them.

2.       In the “Conclusion” session Authors described “Both conditions may ultimately imperil renal function and 556 cardiovascular health.” However, no specific discussion might not be included in the text body.

3.       In authors’ summarized tables, discrepancies could be observed between some results and conclusions. It would be better to remake them carefully for readers.

Minor comments

1.       Please use full terms before usage of any abbreviations including Tables. For example, are “perc” in Table and CKiD in text? What’ different between CKD and CKiD? Also, what do “(1)” and “(2)” mean in “Protein Energy Wasting” and “Frailty-Inflammation phenotype (2)” of Table 1, respectively? ãƒ»ãƒ»ãƒ»and so on. Figure legends should also include full term for each abbreviation. Please clarify them.

2.       Could you add any calculations for children in text or Table? It may be useful for readers.

3.       In the legend of Figure 2, some explanations and abbreviations should be added.

Author Response

Dear reviewer,

Thank you for your review and comments on our manuscript entitled “Malnutrition patterns in children with chronic kidney disease”. We appreciate the possibility to submit a revised version of our manuscript. We carefully revised our manuscript based on the reviews. We think that the changes made have significantly improved the quality of our manuscript. Reviewers comments are in italics, authors’ answers are in plain text, and changes in the manuscript are underlined. A version of the manuscript including the modifications, as well as a clean copy, have been submitted.

Reviewer 2 Report

It is quite comprehensive. The section on growth hormone therapy makes it unique.

Author Response

Dear reviewer,

Thank you for your review and comments on our manuscript entitled  "Malnutrition patterns in children with chronic kidney disease”. We appreciate the possibility to submit a revised version of our manuscript. We carefully revised our manuscript based on the reviews. We think that the changes made have significantly improved the quality of our manuscript. Reviewers comments are in italics, authors’ answers are in plain text, and changes in the manuscript are underlined. A version of the manuscript including the modifications, as well as a clean copy, have been submitted.